# A Rare *MSH2* Variant as a Candidate Marker for Lynch Syndrome II Screening in Tunisia: A Case of Diffuse Gastric Carcinoma

**DOI:** 10.3390/genes13081355

**Published:** 2022-07-28

**Authors:** Maria Kabbage, Jihenne Ben Aissa-Haj, Houcemeddine Othman, Amira Jaballah-Gabteni, Sarra Laarayedh, Sahar Elouej, Mouna Medhioub, Haifa Tounsi Kettiti, Amal Khsiba, Moufida Mahmoudi, Houda BelFekih, Afifa Maaloul, Hassen Touinsi, Lamine Hamzaoui, Emna Chelbi, Sonia Abdelhak, Mohamed Samir Boubaker, Mohamed Mousaddak Azzouz

**Affiliations:** 1Department of Human and Experimental Pathology, Institut Pasteur de Tunis, Tunis 1002, Tunisia; jihenne.benaissa@pasteur.utm.tn (J.B.A.-H.); amira.jaballah@pasteur.utm.tn (A.J.-G.); laarayedhsarra@gmail.com (S.L.); haifa.tounsi@gmail.com (H.T.K.); jheyna@live.com (A.M.); boubaker.samir@yahoo.fr (M.S.B.); 2Laboratory of Biomedical Genomics and Oncogenetics, Institut Pasteur de Tunis, Tunis EL Manar University, Tunis 1002, Tunisia; medhioub.mouna@yahoo.fr (M.M.); amal.khsiba@yahoo.fr (A.K.); mahmoudi.moufida@gmail.com (M.M.); hbelfkih@yahoo.com (H.B.); lamine015@yahoo.fr (L.H.); emnachelbi1@gmail.com (E.C.); sonia.abdelhak@pasteur.utm.tn (S.A.); mm.azzouz@rns.tn (M.M.A.); 3Sydney Brenner Institute for Molecular Bioscience, University of the Witwatersrand, Johannesburg 2000, South Africa; houcemeddine.othman@wits.ac.za; 4Marseille Medical Genetics, Aix Marseille University, INSERM, 13007 Marseille, France; sahar.elouaj@univ-amu.fr; 5Gastroenterology Department, Mohamed Tahar Maamouri Hospital, Nabeul 8000, Tunisia; 6Department of Oncology, Mohamed Tahar Maamouri Hospital, Nabeul 8000, Tunisia; 7Department of Surgery, Mohamed Tahar Maamouri Hospital, Nabeul 8000, Tunisia; hassentouinsi2@yahoo.fr; 8Department of Pathology, Mohamed Tahar Maamouri Hospital, Nabeul 8000, Tunisia

**Keywords:** HDGC, *CDH1*-negative case, lynch syndrome II, genetic screening, target gene sequencing, DNA repair genes, *MSH2*

## Abstract

Several syndromic forms of digestive cancers are known to predispose to early-onset gastric tumors such as Hereditary Diffuse Gastric Cancer (HDGC) and Lynch Syndrome (LS). LSII is an extracolonic cancer syndrome characterized by a tumor spectrum including gastric cancer (GC). In the current work, our main aim was to identify the mutational spectrum underlying the genetic predisposition to diffuse gastric tumors occurring in a Tunisian family suspected of both HDGC and LS II syndromes. We selected the index case “JI-021”, which was a woman diagnosed with a Diffuse Gastric Carcinoma and fulfilling the international guidelines for both HDGC and LSII syndromes. For DNA repair, a custom panel targeting 87 candidate genes recovering the four DNA repair pathways was used. Structural bioinformatics analysis was conducted to predict the effect of the revealed variants on the functional properties of the proteins. DNA repair genes panel screening identified two variants: a rare *MSH2* c.728G>A classified as a variant with uncertain significance (VUS) and a novel *FANCD2* variant c.1879G>T. The structural prediction model of the *MSH2* variant and electrostatic potential calculation showed for the first time that *MSH2* c.728G>A is likely pathogenic and is involved in the MSH2-MLH1 complex stability. It appears to affect the MSH2-MLH1 complex as well as DNA-complex stability. The c.1879G>T *FANCD2* variant was predicted to destabilize the protein structure. Our results showed that the MSH2 p.R243Q variant is likely pathogenic and is involved in the MSH2-MLH1 complex stability, and molecular modeling analysis highlights a putative impact on the binding with MLH1 by disrupting the electrostatic potential, suggesting the revision of its status from VUS to likely pathogenic. This variant seems to be a shared variant in the Mediterranean region. These findings emphasize the importance of testing DNA repair genes for patients diagnosed with diffuse GC with suspicion of LSII and colorectal cancer allowing better clinical surveillance for more personalized medicine.

## 1. Introduction

Gastric carcinoma (GC) is the fifth most common cancer worldwide with approximately one million new cases registered in 2018 (5.7%) with significant geographical distribution variations. It represents the third largest cause of cancer-related death with 783,000 deaths worldwide in 2018, which represents 8.2% of all cancer deaths [1,2]. In Tunisia, as in other countries, the sporadic form of GC is the most frequent, estimated at 637 new cases per year and ranks the 7th among all diagnosed cancers [1]. Inherited syndromic forms of digestive cancers with age occurrence before the age of 50 (gastric and colorectal) seem to be in relatively higher proportions in the Tunisian population, suggesting a genetic susceptibility [3]. However, no epidemiological data are available, and incidence rates of hereditary forms are lacking. Several syndromic forms are known to predispose to gastric tumors such as hereditary diffuse gastric cancer (OMIM: 137215), lynch syndrome (OMIM: 120435), Peutz Jegher Syndrome (OMIM: 175200) and Li Fraumeni syndrome (OMIM: 151623) [4].

Indeed, 3–5% of GCs are caused by autosomal dominant inherited mutations and familial aggregation occurs in approximately 10% of the cases [5]. Hereditary diffuse gastric cancer (HDGC) syndrome is known as the most frequent autosomal dominant form of GC and is a poorly differentiated morphologic type [6]. Approximately 40% of HDGC cases have germline mutations in *CDH1* (E-Cadherin, OMIM: 192090), and more than 100 *CDH1* mutations have been reported in HDGC cases. Lynch Syndrome (LS) (OMIM: 120435), or HNPCC (Hereditary Non Polyposis Colorectal Cancer), is the most prevalent inherited CCR (colorectal cancer) susceptibility syndrome, responsible for up to 6% of all CRCs. It is characterized by point mutations and/or large rearrangements in DNA mismatch repair (MMR) genes resulting in a loss of MMR complex function and microsatellite instability (MSI). The MMR genes associated with this syndrome are: *MLH1* (OMIM: 609310), *MSH2* (OMIM: 609309), *MSH6* (OMIM: 600678) and *PMS2* (OMIM: 600259) [7,8].

In addition, germline deletions in the 3′end of *EPCAM* gene (OMIM: 613244), directly upstream to *MSH2*, have been found to underlie a small proportion of LS cases (<5%) through methylation induced transcriptional silencing of *MSH2* [9].

LS has been subdivided into two subtypes: LSI known as site-specific colonic cancer syndrome, and LSII or extracolonic cancer syndrome (OMIM: 120435) [10]. The former subtype includes 2–3% of all CCRs and is characterized by a susceptibility to colorectal tumors in the absence of diffuse polyposis. The latest subtype is associated with a high risk of various tumors types such as endometrium [11,12,13], gastric [5,14,15,16], breast [17], biliary tract, urinary tract [18,19], small bowel [10], ovarian [20,21,22], brain and skin cancers [23,24,25].

In the context of LS II, MMR genes germline mutation carriers represent 50 to 80% lifetime risk to develop CCR, 40 to 60% risk of developing endometrial cancer (OMIM: 608089) and 13 to 19% risk to develop gastric tumors [26]. Moreover, GC is, among LS carriers, far less common than colorectal at an average older age in western countries [14]. It is well-established that MMR genes dysfunction is linked to LS susceptibility. Indeed, current data assume that *MSH2*, *MLH1*, and *EpCAM* mutation carriers have 13% of cumulative risk incidence of developing gastric tumors at an age of 75 years; however, *MSH6* carriers have only 3% of the risk [27]. Even though GC is part of the LS tumors spectrum the risk of developing GC in LS families is unknown and surveillance strategies for GC in LS are still controversial [14]. To our best knowledge, few studies have investigated the molecular basis and the genetic mutational profile underlying this form of GC. In Tunisia, some molecular and epidemiological studies have been reported in colorectal cases in LS families [3,28,29]. However, no study has investigated the molecular basis of GC cases in LS. Hence, it is of crucial importance to set up oncogenetic counseling based on specific genetic tests adapted to the Tunisian population. This will help in early detection of individuals and families at high risk of developing LSII and will consequently reduce the mortality and morbidity due to the disease. 

Our main objective in the current work was to identify the genetic mutational profile underlying the DGC occurring in a Tunisian family suspected with both HDGC and LSII by using a classical sequencing and target NGS approaches. We used a custom panel targeting 87 candidate genes recovering the four DNA mismatch repair pathway (MMR, BER, NER and HR) known to be involved in DNA repair disorders. The proband diagnosed with a DGC was screened for a repair genes panel after HDGC syndrome exclusion.

## 2. Material and Methods

### 2.1. Patient

This study was conducted according to the declaration of Helsinki and with the approval of the Institutional review board (IRB) of Institut Pasteur de Tunis. Four individuals, belonging to the same large Tunisian family suspected with LSII according to revised Bethesda Guidelines, were investigated after participants wrote informed consent (Figure 1). The index case has been investigated first. She was a woman, “JI-021”, from a consanguineous marriage, fulfilling the first and fifth criteria of the Revised Bethesda Guidelines and the third criteria of IGCLC with no alcohol, smoking consumption habits, spicy foods with low consumption of meats. She was referred first for *CDH1* germline genetic testing to the Gastroenterology Department of Mohamed Tahar Maamouri Medical Hospital in Nabeul, Tunisia for HDGC suspicion since she was diagnosed with diffuse gastric carcinoma (DGC) (T3NxM1) at the antrum at an age of 52. This index case had an aggressive phenotype since diffuse gastric tumors are known to be an aggressive gastric tumor histotype. This patient had peritoneal carcinosis, and she was therefore not operable and the only treatment she had was a palliative chemotherapy. This index case has a healthy 23-year-old daughter, three unaffected brothers (47, 53 and 55 years old) and two affected sisters (29 and 47 years old). Her youngest sister, “JI-021-E”, suffered from primary sterility, developed successively ovarian and gastric carcinomas and was treated by hysterectomy and a total gastrectomy and died at the same age. The other sister, “JI-021-D”, developed a BC, and she died at the same age. The index case’s father, “JI-021-H”, suffered from renal failure and diabetes and died at the age of 65. In addition, this family seems to have a strong history of CCR from the paternal side (five cases) without clear information regarding their relationship with the index case. He had a brother and a sister who died, respectively, at the ages of 50 (Probably from CCR), “JI-021-J”, and 32 years old from unknown causes, and a third sister, “JI-021-F”, who developed a CCR before the age of 50 and died at the age of 65. Her daughter, “JI-021-G”, also developed a CCR at the age of 50. Table 1 shows all the clinicopathological criteria of the index case. 

### 2.2. Sanger Sequencing

#### 2.2.1. *CDH1* Coding Regions Sanger Sequencing

Referring to 2015 IGCLC [30] for the search of *CDH1* and *CTNNA1* [31] germline mutations, a complete sequencing of these two genes was carried out. Primers were designed using Primer Express™ Software version 2.0 Applied Biosystems Thermo Fisher Scientific [32] as shown in Appendix A, used in our previous study [33]. Forward and reverse primers incorporated the extensions 18F tail (ACCGTTAGTTAGCGATTT) and 18R tail (CGGATAGCAAGCTCGT), respectively, at their 5′ ends. Then, generated data were analyzed using SeqScape Version 3.2 (Thermo Fisher, Multiple Life Technologies Corporation, Carlsbad, CA, USA) and BioEdit Sequence Alignment Editor Version 7.2.5.

#### 2.2.2. *MSH2* Sanger Sequencing

We designed two pairs of primers (Forward primer sequence 5′AGAGGAGGAATTCTGATCAC3′, and reverse primer sequence 5′CTGATTCTCCATTTCTGGC3′) to amplify the following region: *MSH2*: exon 4 (NM_000251.2) (Appendix A). PCR reactions were performed on genomic DNAs (gDNAs), following standard protocols, pursued by Sanger sequencing using an automated sequencer (ABI 3500; Applied Biosystems, Foster City, CA, USA) using a cycle sequencing reaction kit (Big Dye Terminator kit, Applied Biosystems, CA, USA). Data were analyzed by BioEdit Sequence Alignment Editor Version 7.2.5.

### 2.3. Large CDH1 Deletions/Duplications Analysis by Multiplex Ligation-Dependent Probe Amplification (MLPA) Assay

The index case’s genomic DNA was tested for copy number changes using Multiplex Ligation Dependent Probe Amplification (MLPA). It was performed using the SALSA P083-D2 *CDH1* MLPA kit (MRC-Holland, Amsterdam, Holland), following manufacturer’s instructions. The kit contains 35 probes for the *CDH1* gene, one upstream flanking probe and one downstream flanking probe. MLPA products were run on the ABI Prism 3730 xl Genetic Analyzer (Applied Biosystems Thermo Fisher, CA, USA) and analyzed with the Peak Scanner™ Software v1.0 (Applied Biosystems Thermo Fisher, CA, USA). 

### 2.4. Next Generation Sequencing

Blood samples have been collected from the index case and her consent relatives and have been sampled in the gastroenterology department. Genomic DNA was isolated from peripheral blood using Qiagen Kit and salting-out, and it was used for the library preparation for NGS. HaloPlex^HS^ assay incorporating molecular barcodes for high-sensitivity sequencing was used as a custom design (HaloPlex^HS^). Using SureDesign (Agilent Technologies Inc., Santa Clara, CA, USA), probes were generated to cover the exons and 15 bp of the surrounding intronic sequences of a total of 87 candidate genes known to be involved in DNA repair disorders [34]. 

Amplicon libraries were prepared, from genomic DNA of “JI-021”, using the HaloPlexHS PCR target enrichment system dedicated to Ion Torrent PGM according to the manufacturer’s recommendations. Massively parallel sequencing was performed on an Ion Torrent PGM (Thermo Fisher Scientific). Raw data generated by the PGM sequencer were analyzed using the VarAFTsoftware version 2.5, which is freely available online (https://varaft.eu/, accessed on 20 July 2022). Genome assembly and nucleotide coordinates were referenced to the GRCh37 version of the human genome. 

### 2.5. Variants Filtering Strategy

We aimed herein to select pathogenic or potentially pathogenic variants associated with the development of LSII by removing neutral variants and sequencing errors. We prioritized rare functional variants (missense, nonsense, splice site variants, and indels) and excluded variants with a Minor Allele Frequency (MAF) > 0.01 in dbSNP137, and 138, in the Exome Variant Server (http://evs.gs.washington.edu/EVS/), accessed on 16 April 2022, 1000 Genomes Project (http://www.1000genomes.org/, accessed on 16 April 2022), or Exome Aggregation Consortium database (ExAC), Cambridge, MA (http://exac.broadinstitute.org, accessed on 16 April 2022), and the genome Aggregation Database (gnomAD) (https://www.ncbi.nlm.nih.gov/bioproject/PRJNA398795, accessed on 16 April 2022).

We also removed all synonymous and homozygous variants considering that LS is an autosomal dominant syndrome. All variants with a Depth ≤ 15 have been removed as well as variants classified as Benign and Likely Benign in ClinVar. Variants predicted with tolerated effect by more than two in silico prediction tools have also been removed. 

Indeed, various in silico prediction tools have been used to assess the functional effect and pathogenicity of the selected variants such as UMD predictor (http://umd-predictor.eu/, accessed on 16 April 2022), Sorting Intolerant From Tolerant (SIFT) (http://sift.jcvi.org/, accessed on 16 April 2022), used to examine the degree of conservation for amino acid residues across species and to find changes in protein structure and function, PolyPhen-2 (http://genetics.bwh.harvard.edu/pph2/, accessed on 16 April 2022), Protein Variation Effect Analyzer (PROVEAN) (http://provean.jcvi.org/, accessed on 16 April 2022), to filter sequence variants to identify nonsynonymous or indel variants that are predicted to be functionally important. Mutation Taster (http://www. mutationtaster.org/, accessed on 16 April 2022), has been used to assess the impact of mutations on protein function and to look at effects on splicing sites, mRNA expression, MAPP-MMR (http://mappmmr.blueankh.com/, accessed on 16 April 2022), to accurate classification of missense variants in MMR genes, FATHMM (http://fathmm.biocompute.org.uk/, accessed on 16 April 2022), to predict the functional consequences of both coding variants (Non-Synonymous single nucleotide variants) and non-coding variants and LRT. Variants not previously reported in healthy controls and classified as pathogenic were evaluated for sequencing depth and visually inspected using the Integrative Genomic Viewer (IGV) before validation by Sanger sequencing.

### 2.6. Immunohistochemical Study

The index case “JI-021” tumor gastric tissue was tested for E-cadherin, MSH2 and MLH1 expression profiles by immunohistochemistry. Briefly, 3–4 µm tissue sections were obtained from FFPE gastric tissue and all immunohistochemistry steps were conducted as preconized in the Novolink MPolymer Detection Systems Kit (Leica Biosystems United States/Biopole, Tunisia). Antigen retrieval was performed by incubation of tissue sections in a 10 mM Sodium Citrate buffer (pH 6.0) (RE7113, Leica Biosystems United States/Biopole, Tunisia) for 20 min in 95 °C heated water.

#### 2.6.1. E-Cadherin

Gastric tissue was incubated forward overnight with anti-E-Cadherin primary mouse monoclonal antibody (NCL-L-E-Cad, clone 36B5, Novocastra TM, Biopole) recognizing the Nt external domain of E-cadherin. The immunostaining reaction was visualized by adding the Diaminobenzidine DAB as a chromogenic substrate using Novolink MPolymer Detection Systems Kit (Biopole).

#### 2.6.2. MSH2 and MLH1

Primary antibodies anti-MLH1 (ES05) and anti-MSH2 (25D12) Leica Biosystems were diluted to 1:50 and were added and incubated for 60 min. The immunostaining reaction was visualized by adding the AEC as a chromogenic substrate (Novolink MPolymer Detection Systems Kit (Biopole)).

### 2.7. Preparing the Structures

#### 2.7.1. The Structure of Two Complexes

MSH2/MSH6 [35] and MSH2/MSH3 [36] were used in our computational study to analyze the effect of the variant R243Q on the MSH2 protein. Missing segments and atoms were added using MODELLER [37]. We also used the structure of the N-terminus domain of the DNA mismatch repair protein MLH1 [38]. The variant at position 243 of MSH2 in both complexes was performed in silico using FoldX [39].

#### 2.7.2. Protein–Protein Docking

We conducted a protein–protein docking experiment using ZDOCK version 3.0 [40] using the structure of MLH1 as a ligand (the smallest partner) and the complex MSH2/MSH6 as a receptor (the largest partner). Input options were kept to their default values. We have, however, restrained the docking interface on MLH1 only to solve exposed residues of the large beta-sheet from the N-terminal domain. We generated 2000 complexes from which we took the best ten according to the Zdock scoring function. We then refined the complexes using sander from AMBER 19 [41] molecular dynamics package by running a restrained minimization consisting of 1000 steps of steepest descent (250 steps) and conjugate gradient (750 steps) algorithms. Restraints were applied to backbone atoms of the complex. A non-bonded cutoff of 12 Angstroms was applied for the refinement.

#### 2.7.3. Electrostatic Potential Calculation

The electrostatic potential calculation [42] has been conducted using the Adaptive Poisson–Boltzmann Solver (APBS version 1.4.1) software package.

The PDB structure of a protein is first converted to the PQR format using the PDB2PQR server [43]. The PARSE force field was used to assign the partial charges to each atom and the ionization state of charged residues was assigned according to the calculation by PROPKA. The electrostatic potential calculation was effected using an implicit solvent model. The internal dielectric constants of the solute and the solvent are fixed to 2.0 and 78.45, respectively, and the temperature is set to 298.15 K. The cutoff for non-bonded interactions was set to 15 Angstroms. Visualization of the electrostatic potential was generated using the Chimera molecular viewer.

#### 2.7.4. Stability Analysis

We estimated the free energy of folding between the wild type and the mutant structure (ΔΔG_Wt–Mut_) using DynaMut [44]. 

## 3. Results

### 3.1. Sanger Sequencing

#### *CDH1* Sanger Sequencing

As a result, we have found no deleterious or potentially deleterious variants in the coding and flanking regions of both *CDH1* and *CTNNA1* genes as it was mentioned in our previous study [33]. This finding suggests other candidate genes/variants predisposing to DGC.

### 3.2. Large CDH1 Gene Deletions/Duplications Analysis by MLPA Assay

Since heterozygous large genes rearrangements are hard to detect by conventional PCR-based sequencing of gDNA, we searched for possible rearrangements of the *CDH1* locus using the MLPA assay [45]. Results were processed by the Coffalyser.Net software (MRC-Holland, Amsterdam, Holland). Our results showed no large rearrangements in the *CDH1* gene among this index case, as it is described in our previous study [33].

### 3.3. E-Cadherin Immunohistochemistry Expression Profile

In order to assess the protein expression level of E-cadherin (*CDH1* gene product) in the index case gastric tumor tissue, we have performed an immunohistochemistry by using a monoclonal antibody directed against the Nt domain of E-cadherin. Our result showed a negative immunostaining expression in tumor cells compared to normal adjacent glandular cells as control (Figure 2A). The adjacent normal gland showed a membranous normal E-cadherin expression pattern. 

### 3.4. Selection of Variants of Interest Detected by Targeted DNA Repair Genes Panel

Besides IGCLC inclusion criteria, this index case also fulfilled Bethesda Guidelines inclusion criteria for testing LS (First, Fourth and Fifth criteria). Hence, this case was a candidate for the screening of a panel of 87 candidate genes. Using SureDesign (Agilent Technologies Inc.), probes were generated to cover exons and 15 bp flanking sequences. The size of the final target region was 251.689 kpb with 33,828 amplicons and the mean coverage was 99.74% of the target region. As a result, we identified 33 variants. Statistical distribution of variants is illustrated in (Figure 3). Exonic variants represented 23 out of 33 variants. Among the 23 exonic variants 20 were novel (Appendix A).

Among the exonic variants, only the rare *MSH2* (rs63751455) variant (with a MAF less than 0.01 in the general population) corresponding to substitutions c.728G>A p.R243Q, classified as a VUS in ClinVar, has been selected for Sanger sequencing validation and for structural model effect prediction analysis. Different in silico prediction tools (15) were used to evaluate the potential functional effect of this variant (Appendix A). The variant is located in a conserved protein domain of the MSH2 protein among several species. Using Mutation taster, this variant has been described as “disease causing” and seems to affect the protein structure. Both Polyphen and SIFT described it as “damaging”. UMD predictor classified it as “pathogenic” and Provean as “deleterious”. In addition, according to the Human Splice Finder tool this variant results in a creation of a new acceptor splicing site and a new silent splicing site “Exonic Splicing Silencers ” (ESS) (wild: 0.55/mutated: 0.62). The MAPP-MMR tool classified it as neutral with a score equal to 2.130. In brief, by using other online supplementary prediction tools (DANN, FATHMM v2.3, Mutation Assessor, LRT…), this variant has been described as “deleterious or pathogenic” by 13 out of 15 tested tools.

This variant has been confirmed for the index case “JI-021” in an heterozygous state and not found in the index case’s relatives (index case’s daughter, brother and her niece). This variant has been forward searched in three other gastric cancer cases, “JI-002”, “JI-030” and “JI-036”, belonging to unrelated families suspected with LS II and fulfilling the first criterion of Bethesda guidelines for the first case “JI-002” and the fifth criterion for both “JI-030” and “JI-036” cases based on their family history of cancer. As a result, it has been found in none of the three screened index cases suspected with LSII. A search of potential linkage disequilibrium of this variant with known pathogenic variants has been performed, and currently no information is available in LDproxy Tool [46].

Additionally, based on structural analysis and DynaMut score on the theoretical functional effect, a novel identified variant on the *FANCD2* gene has been selected for in silico prediction analysis as well. 

### 3.5. MSH2 and MLH1 Immunohistochemistry Staining in the Index Case Tumor Gastric Tissue

The immunohistochemistry analysis of MSH2 and MLH1 proteins in the index case gastric tissue has shown a positive diffuse nuclear immunostaining for MLH1 and nuclear immunostaining of MSH2 in tumor cells and normal adjacent residual glandular cells with loss of MSH2 staining in some tumor cells (incomplete immunostaining) (Figure 2B,C). Since no significant difference was noticed between mutated and wildtype models regarding the putative impact on the collective motion of the residues using MSH2/MSH6, MSH2/MSH3 and MSH2 (monomeric protein) structures, and based on MSH2/MLH1 effect, only MSH2 and MLH1 proteins have been investigated by IHC. Moreover, we no longer have FFPE gastric tissue available for this case since it was a small biopsy.

### 3.6. Molecular Modeling of MSH2 Variant on the MSH2/MSH6/MLH1 Complex

The variant R243Q is located on the connector domain of MSH2 at a loop connecting two alpha-helices (Figure 4A). Most of the atoms of the side chain of arginine are highly packed against other residues forming the core of the domain while the polar guanidine group is exposed and forms a hydrogen bond with the D240 main chain. With such a configuration of R243, we thought first that the variant could impact the conformational function of the connector domain to other parts of the protein. We then run a calculation of the normal modes for the Wild Type (WT) form (R243) and the mutant form (Q243) to investigate the putative impact on the collective motion of the residues using MSH2/MSH6, MSH2/MSH3 and MSH2 (monomeric protein) structures, but no significant difference was noticed We then investigated the possibility that the connector domain is involved in protein–protein interaction with other partners. Indeed, there have been different arguments showing that the connector domain of MSH2 is involved in the interaction with a heterodimer of MLH1/PMS1 [47]. We chose to run a protein–protein docking experiment to predict the ternary complex MSH2/MSH6/MLH1 (Figure 4B). Noting that the docking analysis was conducted only using the wild type structure of MSH2 since the docking score for solving protein–protein complexes are not reliable to determine the stability of a binary association. In other words, the docking was conducted to predict surface patches on MSH2 that are likely to bind MLH1. Of the 10 complexes with the best ZDOCK energy score, six show a binding mode where the connector domain interacts with MLH1, while in the docking solution with the best energy score, R243 is part of the interface.

As the p.R243Q variant involves a substitution of a positively charged residue with a neutral amino acid, we proceeded by calculating the electrostatic potential of MSH2 in both WT and mutant forms. At the site of the variant, the electrostatic potential is majorly electronegative (Figure 5A). We noticed the occurrence of some neutral surface patches and a slightly electropositive region at the exact position of R243 in the wild type form. In the mutant form, however, the electronegativity increases. We verified that the electrostatic potential at the site of the variant is compatible with the interaction interface of MLH1 by calculating the electrostatic potential of the latter protein. Indeed, we found that the protein–protein interface is exclusively electropositive in MLH1 (Figure 5B).

### 3.7. Molecular Modeling of the FANCD2 Novel Variant

Among the other putative deleterious variants, we identified a novel p.A627S variant on FANCD2. FANCD2 forms a protein–protein semi-symmetrical complex with FANCI, which is involved in the physio-pathological mechanism of Fanconi anemia cancer. The substitution is located at the N-terminal end of an alpha helix of the FANCD2 NTD domain. The variant p.A627S is located near a monoubiquitination site on FANCD2 K561 residue (Appendix A). Ubiquitination can activate the Fanconi anemia DNA repair pathway [48]. The variant induces a significant destabilization of the folding energy calculated using DynaMut as well as a perturbation of the local flexibility of the structure [49]. The ΔΔG calculated by DynaMut shows an unfavorable energy of −1 kcal/mol. Moreover, four other tools including EnCoM, mCSM, SDM and DUET also show an unfavorable ΔΔG of −1.8, −2.9 and −2.1, respectively. Therefore, all the tools are in concordance about the destabilizing effect of the p.A627S variant.

## 4. Discussion

Currently, screening for serious diseases such as cancer represents one of the priorities of public health, and therefore of medical research. Indeed, thanks to the development of highly throughput technologies, particularly in the field of genomics, it became easier to elucidate the molecular mechanisms underlying the development and progression of hereditary cancers. In addition, such sequencing approaches contributed significantly to the identification of various genetic actionable variants allowing the development of molecular screening for early diagnosis of complex syndromic cancers, particularly in under-investigated populations such as the North African population.

GC is the second most common extracolonic tumor in LS (HNPCC) [50,51]. Although the risk of extracolonic tumors including gastric seems to be higher in MSH2 compared to MLH1 mutation carriers [52], the association between germline mutational profile and clinical phenotype is generally weak in LS [15]. Few studies have performed molecular comprehensive investigation on patients with gastric tumors even if they are LS mutation carriers or not.

In the current study, we investigated a DGC patient “JI-021” belonging to a Tunisian family from the north-east of Tunisia with suspicion of both HDGC and LSII (according to the tumor’s cluster family history) shedding light on the molecular basis of GC in this region known to have relatively high proportion of digestive cancer syndromes (no epidemiological data available). Based on the International Gastric Cancer Linkage Consortium (IGCLC), this case meets the third criterion for *CDH1* germline screening. As a result, we found no pathogenic variant in the *CDH1* gene coding and exons flanking regions [33]. In addition, no large rearrangements have been identified by MLPA in this gene for this patient [33]. This result is not surprising since the lack of *CDH1* pathogenic variant DGC patients meeting the IGCLC testing criteria has already been reported. Indeed, Jakubowska and colleagues have sequenced the entire coding region of *CDH1* gene in 86 Polish cancer patients from families fulfilling the criteria of HDGC and have found no deleterious mutations in the *CDH1* gene [53].

They have concluded that *CDH1* mutations do not contribute to DGC in Poland. They have, however, some limitations, namely less restrictive criteria of GC patients selection and lack of analysis for large genomic deletions. In fact, our study has covered the *CDH1* entire coding region (16 exons), splice junctions as well as large rearrangements analysis. Nevertheless, we need to investigate the non-coding regions particularly regulatory transcriptional regions and promoter to confirm the definitive HDGC exclusion in this case since its corresponding FFPE tumor tissue harbored a loss of E-cadherin expression in tumor cells comparing to normal adjacent gland (Figure 2A) suggesting a putative *CDH1* gene transcription defect at germinal or somatic level. 

In addition to the IGCLC inclusion criteria, the index case “JI-021” also fulfilled Bethesda Guidelines inclusion criteria for LS testing (1st and 5th criteria). In the current study, our index case gDNA was screened via a custom panel targeting *87* DNA repair genes. Although several studies have used panels covering additional genes other than DNA repair as LS known associated genes or candidate genes, we aimed to focus on a panel covering the four DNA repair pathways in order to identify new candidate DNA repair genes/variants predisposing to such cancer syndrome. As a result, we have identified the MSH2 (c.728G>A p.R243Q) variant in the index case “JI-021”, but not in her asymptomatic relatives (with available DNAs). According to our best knowledge, this is the first study investigating the genetic predisposition of GC case in the context of LSII in Tunisia, and very few studies have reported such an investigation worldwide [14,15].

Indeed, Capelle and colleagues have reported 2014 mutations identified in the MMR genes as part of LS in 236 Dutch families, and among them GC was diagnosed in 32 subjects (1.6%), including 22 (69%) with family history of GC. The risk of developing GC was rated at 4.8% for carriers of the *MLH1* gene mutation and at 9% for carriers of the *MSH2* mutation. However, among 378 *MSH6* identified mutations, no carrier had GC [14].

Nevertheless, with the emergence of precision oncology and development of germline panels, several studies have investigated LS susceptibility genes in individuals with cancers and individuals at risk to develop hereditary cancers.

It is well known that LS is histologically and genetically a complex disease with heterogeneous molecular background. Indeed, it seems that different families with the same LS inclusion criteria could have different mutational profiles in newly identified susceptibility genes not initially known to be associated with LS. These identified mutations/variants appear to have low frequencies among screened cases and families. This reported result is in concordance with our finding since our *MSH2* c.728G>A identified variant has been found only in one case among the three screened GC cases belonging to LSII suspected families. While several other works have investigated the molecular background of CCR in the context of LS worldwide [54,55,56,57,58,59,60,61], only three studies have been conducted in Tunisia in this field. Indeed, Moussa and colleagues have reported germline mutations in MMR genes in 11/31 families suspected for LS (six *MSH2* and five *MLH1* mutations). Among the six *MSH2* identified variants, our *MSH2* c.728G>A variant has been found in only two probands (52 and 38 years old) belonging to two families sharing the fifth Bethesda guidelines selection criterion and the same mutational profile as well as a common newly identified *MSH2* c.1413dupA mutation. These authors reported that *MSH2* c.728G>A (p.R243Q) variant was predicted as probably non-pathogenic since the substituted Arginine residue was relatively little conserved across phylogeny with weak physicochemical difference between arginine and glutamine. However, this claim does not consider the complexity of domain–domain cooperatively, the protein–protein interactions and the fact that Arg residues are positively charged and Gln is neutral. They have found this variant in two families, the first one with five cases of CCR, the second with two cases of CCR. Both families shared the fifth Bethesda Guidelines criterion and two mutations “*MSH2* c.728G>A” and “c.1413dupA”. Compared to our results, the proband “JI-021” carrying the same *MSH2* c.728G>A (p.R243Q) variant shared the same age of GC onset (52) and the same Bethesda criteria (1st and 5th) with the two lastly reported Tunisian families. These homologies suggest that the *MSH2* c.728G>A (p.R243Q) variant might be a LS/LSII predisposing variant in Tunisian population; however, supplementary molecular investigation among a larger cohort is needed to confirm this hypothesis. Intriguingly, Moussa and colleagues noted that the age of cancer onset as well as the tumors spectrum were similar in families with and without MMR germline mutations, which is in contradiction with other studies conducted in other populations [3]. More recently, next generation sequencing of MMR genes, *POLE* and *POLD1* was performed, by Ben Sghaier and colleagues, to identify the genetic mechanisms underlying CCR in 24 Tunisian probands. As results, they identified, in six cases, five germline variants in *MLH1*, a somatic pathogenic variant in *MSH2* (c.2557G>T) and a germline variant in *MSH2* (c.1413 dupA), which was previously reported in the study of Ben Sghaier and colleagues [28]. In addition, a team in our lab have identified in a LS family with discordant twins, an *MSH2* pathogenic mutation (c.1552C>T;p.Q518X) previously reported only in a Portuguese LS family and shared with all investigated family members suggesting that this variant is a causal mutation [29]. In addition, an Algerian recent study have screened *MLH1*, *MSH2* and *MSH6* genes among 21 families from East of Algeria and have identified the *MSH2* (c.728G>A) mutation in two cases (44 and 38 years old) with CCR belonging to the same family with colon cancer family history [62]. The *MSH2* (c.728G>A) variant seems to be identified only in Mediterranean countries. These findings support the fact that this variant might be an LSII rather than LSI predisposing variants in Tunisian population and appears to be a shared variant in the Mediterranean region. Indeed, previous work in our lab has reported that the Tunisian population shares founder variants with other North African and Middle Eastern populations for 43 inherited conditions [63]. Additionally, other variants appeared to be specific to the Tunisian population and shared by other populations as the case for 11-β hydroxylase deficiency as well as breast and ovarian hereditary cancer [64,65,66,67]. Such a phenomenon could certainly be explained by all migratory waves that occurred with the colonial period resulting in the occurrence of new mutations with an important impact on the genetic diversity of the Tunisian population. All these data highlight the importance of founder and shared mutations in decision making tools for diagnosis and prevention of diseases in North Africa, Middle East and migrant populations living in Europe or Mediterranean region [63].

Moreover, our index case harboring the *MSH2* variant (c.728G>A) has an aggressive phenotype since DG tumors are known to be an aggressive gastric tumor histotype. This patient had peritoneal carcinosis, and she was therefore not operable and the only treatment she had was a palliative chemotherapy. The MSH2 and MLH1 immunohistochemistry analysis in index case gastric tumor tissue has shown a positive nuclear immunostaining in tumor cells for MLH1 and an incomplete nuclear immunostaining for MSH2 protein (Figure 2B,C). This result is in good agreement with in silico prediction analyses of MSH2 variant since the predicted effect involves the MSH2/MLH1 complex destabilization, which has no effect on the protein tissue expression and detection. According to our best knowledge, this study is one of the rare investigations of the MMR proteins expression profile in DGC tissue in a LSII suspected family. The MMR protein expression analysis by IHC is usually performed in CRC tissues according to international recommendation for LS diagnosis in routine. However, few studies have investigated MMR status in tumor tissues corresponding to other LS-associated cancers such as GC. Indeed, it has been reported that MMR deficiency seems to be much more frequent in CRC tumors compared to gastric tumors belonging to LS families [68]. These authors have reported that among 45 LS confirmed families 31 cases had GC and only four gastric cases were MMR deficient versus 27 MMR proficient cases. 

Shedding light on the molecular pathogenesis of GC in LS settings, gastric tumors have been included in the modified Amsterdam criteria II, however the clinical consensus criteria for gastric tumors diagnosis in LS suspected cases/families still remain unclear and germline mutations search as well as MSI status are not yet clearly indicated for gastric tumors. Actually there is growing evidence that such tumors from MMR gene germline mutation carriers are part of LS tumor spectrum and relative risk of GC in LS mutation carriers is reported to be higher by 4–19 compared to the general population in western countries [50,51,69] at least by 2-fold in endemic areas in Asia [16] as our investigated region. In fact, taking into account the gene–environment interactions, our investigated family came from an endemic region of Tunisia with relatively high incidence of digestive cancers known with significant exposure of residents to pesticides and characterized by the spread of water with high levels of toxic chemicals; however, no epidemiological or environmental studies have been carried out in this region. These findings highlight the crucial importance of early screening of individuals at high risk of developing such tumors, particularly for these under-investigated in Tunisia.

In the context of LS, this variant is classified in class 3 (Variant with uncertain significance ‘VUS’) by the LOVD database according to the InSiGHT classifications (http://insight-database.org/ accessed on 20 July 2022) and no functional studies on its effect on MSH2 protein structure, function or its interactions with other partners are available. Hence, we performed several in silico prediction analyses using online prediction tools and databases (15 prediction tools), as well as structural modeling of MSH2 known complexes.

The variant p.R243Q as part of the interface between MSH2 and MLH1 appears to be involved in the destabilization of the protein–protein interaction. In their work, Groothuizen et al. [70] suggest the proximity of the connector domain to the MutL binding surface with MutS. Nevertheless, the structure of MutS deviates significantly from that of MSH2 to be able to infer any homology-based conclusions about the contribution of the residue at the 243rd position, hence the utility of our protein–protein predicted complex.

The change in the electrostatic interaction surface of *MSH2* upon variance might result in different association equilibrium with MLH1. The DNA repair mechanism is a multistep process that includes cooperation between different proteins. Electrostatic forces that act at distances of 5–10 Å can be affected by the p.R243Q variant [71]. More recently, it has been shown that the electrostatic potential disruption between MLH1 and MSH2 can lead to the accumulation of DNA errors and an impact on the protein–protein complex [72]. The electrostatic properties of MLH1 seem to be highly regulated via acetylation/deacetylation by the Histone deacetylase 6 (HDAC6). Moreover, the results from [72] also imply that a dynamic of association/dissociation between MutLα, MutSα and HDAC6 is required for DNA repair. In this regard, electrostatic forces, disrupted by the p.R243Q, would play an important role and could result in a significant effect. 

Beyond the *MSH2* variant, another putative probably deleterious variant in the *FANCD2* gene (p.A627S) was identified. *FANCD2* is required for maintenance of chromosomal stability. It also promotes accurate and efficient pairing of homologs during meiosis. It is involved in the repair of DNA double-strand breaks, both by homologous recombination and single-strand annealing. It may participate in S phase and G2 phase checkpoint activation upon DNA damage playing a role in preventing breakage. This gene encodes the protein for complementation group D2. 

The A627 amino acid is located at the opposite helices facing the protein–protein interface with FANCI that also contains an ubiquitination site at residue K561. *FANCD2* ubiquitination stabilizes a conformational change of the protein–protein complex [48] that is required to increase the affinity for the double stranded DNA [49]. The perturbation of the flexibility by p.A627S variant of *FANCD2* may result in two effects. First, the formation of the complex *FANCD2* and the ubiquitin-conjugating enzyme E2-UBE2T would have to overcome the energy barrier induced by p.A627S local destabilization since the variation occurs at a close distance to K561. The second effect may result in the impairment of the association between *FANCD2* and *FANCI* induced by destabilization of the local interface near the variant’s site. Moreover, since monoubiquitination occurs on the evolutionarily conserved lysine residues, variants affecting FANCD2 at K561 residue seem to result in major molecular defects in response to DNA damage agents in the Fanconi Anemia cells [73,74].

Our index case comes from a consanguineous marriage with accumulation of two potentially deleterious variants besides potential deficiency in the cell adhesion system, suggesting a putative increasing risk of several cancers in this family. Unfortunately, we were not able to screen the index case’s brother with CCR or her two affected sisters (breast and ovarian cancers) for this variant since they were dead. Susceptibility to cancer in offspring of consanguineous marriages has been largely studied. For countries such as Tunisia with common consanguineous marriage, the association between consanguinity and mortality due to cancers is highly important for public health programs [75,76,77]. Moreover, it has been reported that consanguinity and inbreeding plausibly led to the accumulation of population-specific founder pathogenic/or likely pathogenic sequence variants (PSVs) [78]. In fact, two reported studies have observed that inbreeding and ROH result in an increased risk of CCR [79] and developing leukemia, lymphoma, colorectal and prostate cancer [80]. Supplementary genetic screenings of this variant in larger cohorts in this region as in Tunisia are needed to confirm these hypotheses.

## 5. Conclusions

Our findings highlight the genetic heterogeneity of cancer predisposition in Tunisia and the importance of the use of target NGS to identify clinically actionable genetic variants for disease management. To our best knowledge, no study has investigated the molecular basis of GC in LS context in either Tunisia or North Africa. Our findings suggest that the *MSH2* c.728G>A (p.R243Q) rare variant could represent a candidate marker for LS II screening. We suggested this variant be shared with Mediterranean countries. Nevertheless, genetic screening of a larger cohort of GC is needed to confirm its association with LSII syndrome in Tunisia. Our structural model prediction showed that this variant is likely pathogenic and is involved in the MSH2-MLH1 complex stability suggesting the revision of its status from VUS to likely pathogenic. The novel *FANCD2* variant appears to have putative functional effects as well. The identification of such likely deleterious variants will help in the setting up of specific clinical surveillance protocol for individuals at high risk of developing this disease in terms of yearly endoscopy and colonoscopy as well as a mammography biannually for women at risk. We acknowledge however the importance of functional validation for our results of both identified variants (the VUS and the novel *FANCD2* variant) which would be the next focus of our study.

## Figures and Tables

**Figure 1 genes-13-01355-f001:**
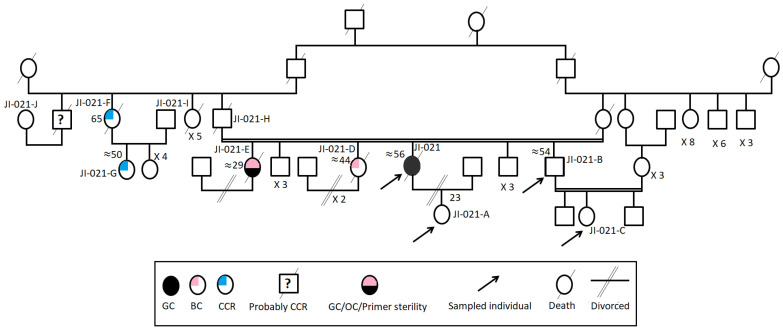
The familial pedigree of the index case “JI-021”. It included gastric, colon, breast and ovarian cancers. It showed a tumor spectrum typical of LS II form.

**Figure 2 genes-13-01355-f002:**
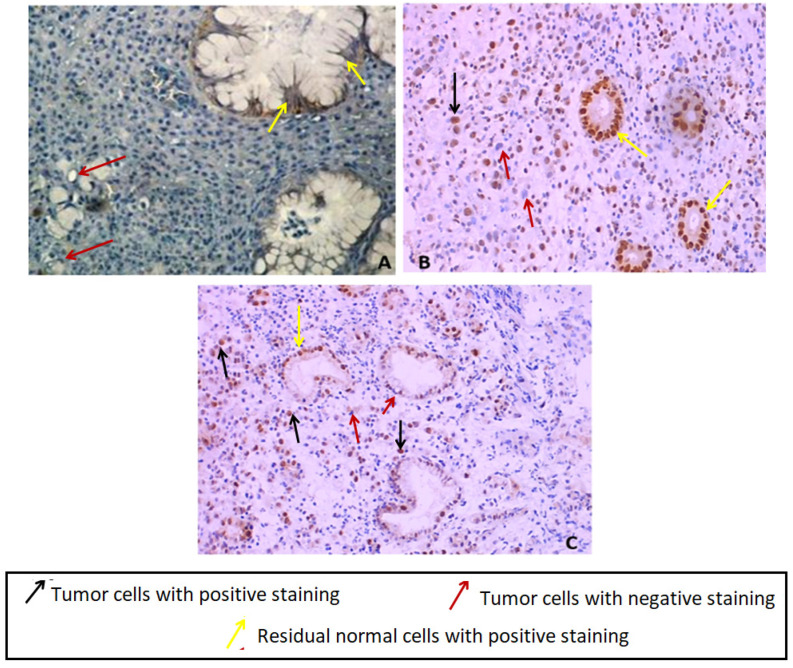
(**A**) Immunostaining of E-Cadherin in index case GC tissues showing a loss of E-cadherin in tumor cells compared to normal glandular adjacent cells showing positive membranous staining (×400). (**B**) Nuclear positive immunostaining of MLH1 in index case GC tissue (×400). (**C**) Uncomplete nuclear positive immunostaining of MSH2 in index case GC tissue (×200) original magnification.

**Figure 3 genes-13-01355-f003:**
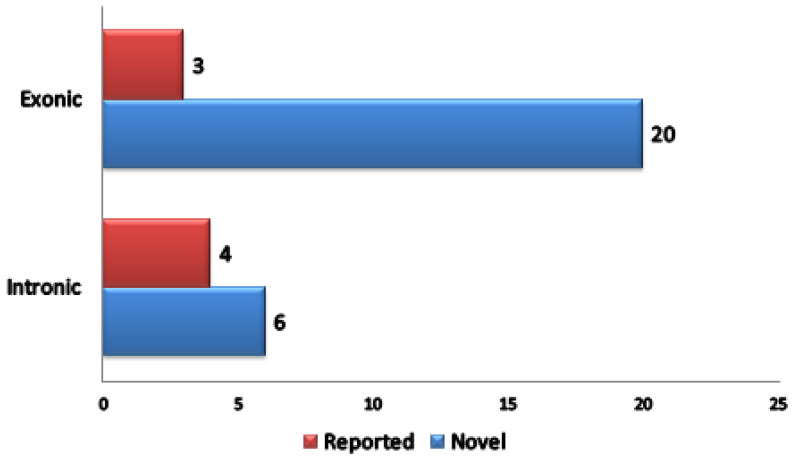
Statistical distribution of variants identified after filtering strategy.

**Figure 4 genes-13-01355-f004:**
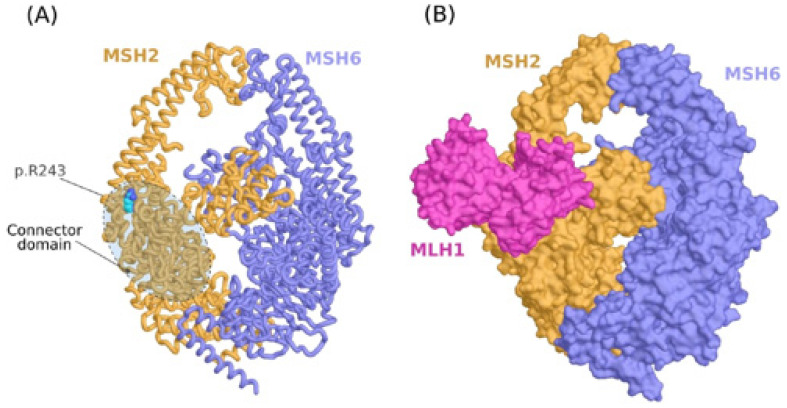
Molecular modeling analysis of p.R243Q variant. (**A**) Molecular model of the MSH2/MSH6 complex showing the location of the variant on the connector domain of MSH2. (**B**) Ternary complex of MSH2/MSH6/MLH1 predicted by protein-protein docking.

**Figure 5 genes-13-01355-f005:**
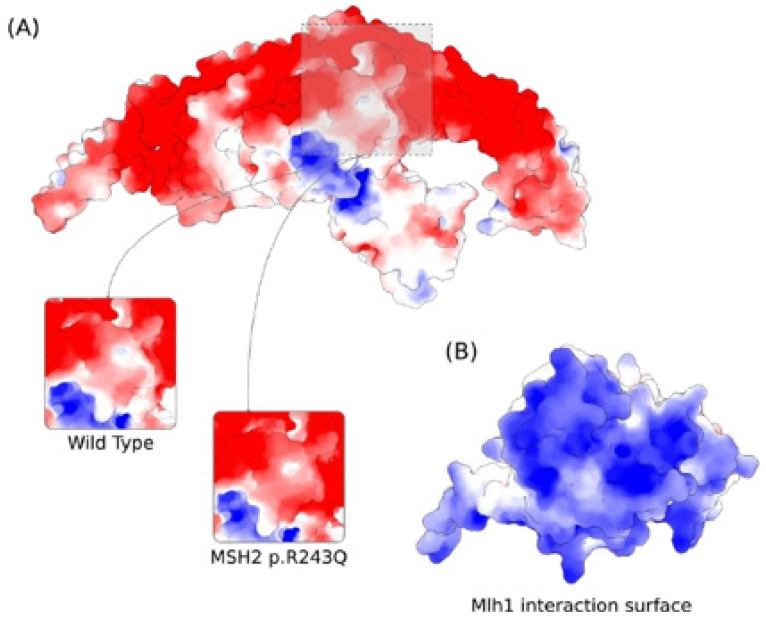
Predicted effect of MSH2 p.R243Q variant on the electrostatic properties of MSH2 calculated by the Adaptive Poisson–Boltzmann Solver. Red and Blue colors correspond to potentials of electronegative and electropositive intensities, respectively. The maximum intensity values are −5 and +5 kb T e^−1^. (**A**) Electrostatic potential of MSH2 showing the variation at the site of the mutation for the wild type and p.R243Q forms. (**B**) Electrostatic potential of MLH1 at the interaction surface with MSH2.

**Table 1 genes-13-01355-t001:** Description of clinicopathological features and family history of the index case.

Case	JI-021
Diagnosis Age/Sex	53/F
Family History First degree relatives Second degree relatives	Colorectal/Ovarian/Breast/Gastric Carcinomas
IGCLC criteria *	Third
Revised Bethesda Guidelines **+**	First, fifth
HP	Yes
Lauren Classification	Diffuse
Location	Antrum
TNM	T3NxM1
Survival	Died after three years of diagnostic (56 years old)

* (1) Colorectal or uterine cancer diagnosed in a patient who is less than 50 years of age. (2) Presence of synchronous, metachronous colorectal, or other HNPCC-associated tumors, regardless of age. (3) Colorectal cancer with the MSI-H. Histology diagnosed in a patient who is less than 60 years of age. (4) Colorectal cancer diagnosed in one or more first-degree relatives with an HNPCC-related tumor, with one of the cancers being diagnosed less than 50 years of age. (5) Colorectal cancer diagnosed in two or more first- or second-degree relatives with HNPCC-related tumors, regardless of age. NB: Hereditary nonpolyposis colorectal cancer (HNPCC)-related tumors include colorectal, endometrial, stomach, ovarian, pancreas, ureter and renal pelvis, biliary tract, and brain (usually glioblastoma as seen in Turcot syndrome) tumors, sebaceous gland adenomas and keratoacanthomas in Muir–Torre syndrome, and carcinoma of the small bowel. **+** (1) Two or more GC cases regardless of age, at least one confirmed DGC, in first-degree and second-degree relatives. (2) One case of DGC before 40 years old. (3) Personal or familial history of DGC and LBC, one diagnosed before 50 years old.

## Data Availability

All data generated or analyzed during this study are included in this published article and its additional files.

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
