# Peer review of "A Rare *MSH2* Variant as a Candidate Marker for Lynch Syndrome II Screening in Tunisia: A Case of Diffuse Gastric Carcinoma"

_genes, 2022, doi:10.3390/genes13081355_

Round 1

Reviewer 1 Report

This manuscript describes that germline MSH2 p.R243Q variant is involved in Lynch Syndrome II.  This is interesting results.

1.  MSH2 p.R243Q variant was detected in proband (JI-021), and not in unaffected relatives (JI-021-A, JI-021-B, and JI-021-C).  It is insufficient to describe that this variant is involved in Lynch Syndrome II.  The author should examine this variant on DNA extracted from blood or non-tumor tissue samples of affected relatives (JI-021-G, JI-021-D, JI-021-E, and/or JI-021-F), and should show uncomplete nuclear expression of MSH2 in tumor tissue from cases with this variant.

2.  Molecular modeling analysis showed that MSH2 p.R243Q variant was involved in MSH2-MLH1 complex stability.  However, it is unclear to what extent this variant affects stability.  The authors should evaluate the stability of the MSH2-MLH1 complex by in vitro assay.

Several typos are found:

1.  Change “Table 2” to “Table 1” on line 143.

2.  Change “JIG-21G” to “JI-021-G” in Figure 1.

Author Response

Dear Sir,

We would like to thank you for accepting the review of our manuscript and for your pertinent comments and suggestions. Please find our answers point by point:

  1. The MSH2 p.R243Q variant has been identified in proband JI-021 but not in consent unaffected relatives (JI-021-A, JI-021-B, and JI-021-C). Regarding the investigation of this variant on DNA extracted from blood or non-tumor tissue samples of affected relatives (JI-021-G, JI-021-D, JI-021-E, and/or JI-021-F), unfortunately, we have no access to their samples since -JI-021-G has refused to consent to be sampled.

         -JI-021-D  developed breast cancer at the age of 44 and died at the same age and we have had no access to her tissue samples/Blood since she died.

         -JI-021-E  developed ovarian and gastric carcinomas at the age of 29 and died at the same age so we can not access her samples.

         -JI-021-F developed a CCR before the age of 50 and died at the age of 65 so we can not access her samples.

Regarding the MSH2 protein immunostaining in proband harboring the variant (JI-021), it has been performed in gastric tumor tissue and has shown nuclear immunostaining of MSH2 in tumor cells and normal adjacent residual glandular cells with loss of immunostaining in some tumor cells which is considered as an incomplete nuclear expression in tumor cells of this patient (Lines from 366 to 375 in results section 3.4 and discussion section from line 570 to line 572).

  1. The MSH2 p.R243Q  variant affects the electrostatic properties of the protein-protein interface between MSH2 and MLH1. By comparing the electrostatic map of the wild type and the mutant forms, we detected a shift toward lesser electronegativity at the position of the residue 243. Electrostatic focus are known to act at larger distance ranges than any other non covalent bonds. We cited the study by Zhang et al (20189, PMID 30770470) to highlight that electrostatic forces are necessary to regulate the supra complex that acts upon DNA damaging signals. Such an effect was extensively discussed in lines 630-640. The functional analysis would be an ideal choice to demonstrate the protein-protein features of MSH2-MLH1 interaction. Experimental validation is beyond the scope of our work. We acknowledge however the importance of functional validation for our results which would be the next focus of our study (Informations were added in the manuscript at the end of the conclusion, lines from 690-692).

Reviewer 2 Report

In this manuscript, the authors aim to identify the mutational spectrum underlying the genetic predisposition to diffuse gastric tumors occurring in a Tunisian family suspected of both HDGC and LS II syndromes. This manuscript is overall clearly written. However, I do have a minor comment for the authors.

1. Figures 1, 2, 3, 4 and 5 are very unclear, higher quality figures should be provided instead.

Author Response

Dear sir, 

We would like to thank you for accepting the review of our manuscript and for your comments. 

Please note that figures 1 and 2 have been re-uploaded with a higher resolution and more detailed legend has been added to figure 2.

Kind regards. 

Reviewer 3 Report

Kabbaage et al., provide an extensive report examining an MSH2 variant as a putative Lynch syndrome (LS) causing pathogenic mutation. The report is informative and is of interest to both basic science and clinical researchers. My concerns and suggestions to improve the manuscript are as follows. 

  1. The use of the term “mutational spectrum” is a bit misleading. The authors are just examining one/two candidate gene variants and not a spectrum of mutational changes. 

  1. Page 3, line 12: I think the authors meant four DNA repair pathways. Kindly verify. 

  1. Fig.2 needs to be better annotated clearly showing the normal and tumor regions and identifying them in the tissue. 

  1. It is not clear to me why the authors picked only the FANCD2 variant to discuss further while mentioning that 32 other variants were also reported from this patient. (Page 3, line # 326). Kindly explain. 

  1. The authors did not report on the MSI status of the tumor. It is essential to report on the status of MSI for LS diagnosis. 

  1. Page 11, line # 368: Kindly revisit ref# 47. I think it is misplaced.  

  1. Authors did not properly cite the previous biochemical work showing involvement of connector domain in MSH2-MLH1 interaction. The work of PMID: 20080788 and PMID: 26163658 are both important in this context. The latter one has been cited but not in the relevant place. Warren et al., 2006 also mention three loops that can potentially serve as protein-protein binding sites on MSH2 connector domain. One of the three loops has the AA under study (R243). It is cited by the authors but not in this context. 

  1. Are there any known LS pathogenic mutations found in the connector domain? Kindly comment. 

  1. The authors kind of stopped short in explaining the effect of a R>Q substitution using the docking experiment. Did the docking experiment with the R>Q substitution show loss of MLH1 binding to MSH2? Kindly rephrase on page 11, line # 384-385. 

  1. Legends for Fig. 2 and Fig. 3 are exactly the same. Kindly correct. 

  1. Did the authors stain for FANCD2? 

  1. Page 13, line # 419 has a typo in “farmer”. 

  1. Why do the authors think that the MSH2 variant is driving cancer? And not the FANCD2 one? This is specifically important because the FANCD2 variant has not been adjudged to be a neutral one. The authors mention this to be a destabilizing variant. Kindly explain. 

  1. Did the authors use any mismatch repair specific in silico tools like MAPP-MMR and PON-MMR to predict functional status? 

  1. Maybe it is beneficial to discuss the MSH2 variant in the context of its presence/absence in population database such as EXAC/gNOMAD as reported for some variants in the supp. Table 2. 

  1. The discussion section is way too extensive. It is informative, no doubt, but maybe it’s better to shorten it to improve readability. Kindly consider. 

  1. It is important to highlight that for a subsequent classification of the MSH2 variant, it needs to be assessed by functional studies either in vitro (PMID: 30504929) or in human cells (PMID: 31237724). 

Author Response

Dear Sir,

We would like to thank you for accepting the review of our manuscript and for your pertinent comments and suggestions. Please find our answers point by point:

Kabbaage et al., provide an extensive report examining an MSH2 variant as a putative Lynch syndrome (LS) causing pathogenic mutation. The report is informative and is of interest to both basic science and clinical researchers. My concerns and suggestions to improve the manuscript are as follows. 

The use of the term “mutational spectrum” is a bit misleading. The authors are just examining one/two candidate gene variants and not a spectrum of mutational changes. 

In our study, we used the term mutational spectrum since we aimed to identify the mutational spectrum of a patient with suspected LSII using the target gene sequencing of MMR genes containing 87 genes. As a result, we identified only two variants in  "MSH2 and FANCD2" genes following the variant filtering methodology (methods section 2.4.1 from lines 209 to 245). These two variants were explored by molecular modeling afterwards.

Page 3, line 12: I think the authors meant four DNA repair pathways. Kindly verify. 

As recommended, we changed DNA repair “systems” in lines 36 (abstract section) and line 472 (discussion section) by DNA repair “pathway”. Also, we changed  DNA repair “genes” in line 114 (introduction section) by DNA repair “pathway”.

Fig.2 needs to be better annotated clearly showing the normal and tumor regions and identifying them in the tissue. 

 Figure 2 has been annotated as recommended to distinguish normal from tumor cells in the tissues. Besides, we added yellow arrows indicating  the residual gland normal cells with positive immunostaining (for the three markers). Black arrows show tumor cells with positive staining (for MSH2 and MLH1). Red arrows show negative staining in tumor cells (for three markers).

It is not clear to me why the authors picked only the FANCD2 variant to discuss further while mentioning that 32 other variants were also reported from this patient. (Page 3, line # 326). Kindly explain. 

After applying the filtering strategy mentioned in our manuscript from line 209 to line 249, we found 33 variants (A list of variants was submitted as a supplementary file: Table S2). Among 33 variants, we selected two for the docking study since they appeared to be the most relevant candidate regarding the in silico tool prediction :

*MSH2 p.R243Q  variant  which is the only VUS identified  was described by 13 out of 15 in silico prediction tools (described in methods section) as a deleterious or pathogenic variant.

*Novel FANCD2 variant which was selected as it is predicted as pathogenic by 2 out of 15 prediction tools (mentioned above). In addition, based on structural analysis and DynaMut, EnCoM, mCSM, SDM and DUET scores on the theoretical functional effect of the 33 selected variants after filtering methodology, this novel variant has been selected for in silico prediction analysis as MSH2 variant.

The authors did not report on the MSI status of the tumor. It is essential to report on the status of MSI for LS diagnosis. 

We believe that MSI status of the Colorectal tumor is highly recommended for LS diagnosis. However, the clinical consensus criteria for gastric tumor diagnosis in LS suspected cases/families remain still unclear and germline mutation search, as well as MSI status, are not yet clearly indicated for gastric tumor. In addition, the index case tissue (small biopsy) was insufficient to have enough DNA tumor material to perform MSI analysis and the patient died. So it was impossible to do an additional endoscopy.

Page 11, line # 368: Kindly revisit ref# 47. I think it is misplaced.  

We would like to thank you for your comment, Ref 47 has been replaced by the right reference :https://doi.org/10.1038/s41594-018-0092-y . Ref titled :” Identification of Exo1-Msh2 interaction motifs in DNA mismatch repair and new Msh2-binding partners”.

Authors did not properly cite the previous biochemical work showing involvement of the connector domain in MSH2-MLH1 interaction. The work of PMID: 20080788 and PMID: 26163658 are both important in this context. The latter one has been cited but not in the relevant place. Warren et al., 2006 also mention three loops that can potentially serve as protein-protein binding sites on MSH2 connector domain. One of the three loops has the AA under study (R243). It is cited by the authors but not in this context. 

We thank the reviewer for pointing out these issues. First, we believe that the study referenced by the reviewer as "Warren et al 2006" is in fact the study by "Joshua J Warren et al 2007" (PMID 17531815). We have cited the paper at the start of section 2.6.1 as a reference for the structure used to conduct the in silico analysis for the complex MSH2/MSH6. 

To better put reference [79] into context, line 584-585 has changed from  

"The crystal structure, presented by [79], suggests the proximity of the connector domain to the MutL binding surface with MutS" to become "In their work, Groothuizen et al [79] suggest the proximity of the connector domain to the MutL binding surface with MutS".

Are there any known LS pathogenic mutations found in the connector domain? Kindly comment.

According to the litterature, it has been reported that several MSH2 variants including  p.L187P,  p.L173P and p.G164R have been identified  in the connector domain (from AA 125 to 299) in HNPCC patients and authors highlighted their pathogenic effect by functional assays (Ref: https://doi.org/10.1053/j.gastro.2006.08.044).

The authors kind of stopped short in explaining the effect of a R>Q substitution using the docking experiment. Did the docking experiment with the R>Q substitution show loss of MLH1 binding to MSH2? Kindly rephrase on page 11, line # 384-385. 

We believe that the reviewer is referring to lines 365-372 in the submitted paper for the docking experiment rather than lines 384-385 which report on the electrostatic calculation. The docking was conducted to predict protein surface patches with which MSH2 is likely interacting with MLH1. In this regard, upon variant, the docking scoring function (and scoring functions in general) is not reliable in telling if the interaction will be lost or held. Rather from this experiment we intended to show that the position of R243 on the MSH2 protein is likely involved in the formation of a binary protein-protein interaction complex with MLH1 as partner. 

This text was added in the results section from lines 377-380: “Noting that the docking analysis was conducted only using the wild type structure of MSH2 as the docking score for solving protein-protein complexes is not reliable to determine the stability of  a binary association. In other words, the docking was conducted to predict surface patches on MSH2 that are likely to bind MLH1”. 

Legends for Fig. 2 and Fig. 3 are exactly the same. Kindly correct. 

First, we would like to inform you that Figures 4 and 5 had the same legend. Thank you for noticing this. Legend for Figure 5 has been corrected in the revised version of the manuscript (results section 3.5) as follow:

Figure 5: Predicted effect of MSH2 p.R243Q mutation on the electrostatic properties of MSH2 calculated by the Adaptive Poisson-Boltzmann Solver. Red and Blue colors correspond to potentials of electronegative and electropositive intensities respectively. The maximum intensity values are -5 and +5 kb T e-1. (A) Electrostatic potential of MSH2 showing the variation at the site of the mutation for the wild type and p.R243Q forms. (B) Electrostatic potential of MLH1 at the interaction surface with MSH2.

Did the authors stain for FANCD2? 

No, we haven't done an IHC for this protein since we haven't enough tumor tissues to do so.

Page 13, line # 419 has a typo in “farmer”. 

The paragraph (from lines 436 to 444 in the discussion section) has been removed in order to reduce the discussion section as recommended. 

Why do the authors think that the MSH2 variant is driving cancer?

This gene is known to be one of the MMR pathway and represents one of the most important candidate genes associated to Lynch syndrom. Besides, this rare variant, classified as a VUS,  has been identified only in cases with colorectal cancers and 13  in silico prediction tools have predicted it as deleterious or pathogenic. Also the MSH2-MLH1 complex is a key step in the repair signaling as their complex allows their nuclear translocation for accomplishing their role. All these arguments highlight the importance of investigating the effect of  this variant on the putative role in tumor cells development.

And not the FANCD2 one? This is specifically important because the FANCD2 variant has not been adjudged to be a neutral one. The authors mention this to be a destabilizing variant. Kindly explain. 

For FANCD2, we performed the mapping of the mutation in conjunction with the stability impact prediction using DynaMut, EnCoM, mCSM, SDM and DUET all of whom are in concordance that the variant is probably deleterious in terms of protein stability. With such an insight, we explored two possible scenarios via which the deleterious effect could be established at the structural level, i.e. the interaction with ubiquitin-conjugating enzymes E2-UBE2T and FANCI affected by the perturbation of their corresponding protein-protein interfaces.

Did the authors use any mismatch repair specific in silico tools like MAPP-MMR and PON-MMR to predict functional status? 

In silico prediction tools used in the current study were detailed in the section methodology from lines 209 to 249. Results of this prediction were resumed briefly from lines 600 to 605. The MSH2 variant has an MAPP-MMR Score of 2.130 which means it has a neutral effect.

Maybe it is beneficial to discuss the MSH2 variant in the context of its presence/absence in population database such as EXAC/gNOMAD as reported for some variants in the supp. Table 2. 

Taking into account your pertinent comment, we added this paragraph at the results section section 3.3  in lines from 340 to 351: The MSH2 c.728G>A variant (rs63751455) is a VUS assigned to "Uncertain significance" annotation in ClinVar resulting in the substitution of Arginine to Glutamine at the 243 position of protein sequence. No information regarding linkage disequilibrium related to this variation is currently available from LDproxy Tool. It is a rare variant with a MAF less than 0.01 in the general population. Different in silico prediction tools (15) were used to evaluate the potential functional effect of this variant (Table S2). The variant is located in a conserved protein domain of the MSH2 protein among several species. Using Mutation taster this variant has been described as “Disease causing” and seems to affect the protein structure. Both Polyphen and SIFT described it as “Damaging”. UMD predictor classified it as “pathogenic” and Provean as “deleterious”. In addition, according to Human Splice Finder tool (http://www.umd.be/HSF/) this variant results in a creation of a new acceptor splicing site and a new silent splicing site "Exonic Splicing Silencers ”(ESS) (wild: 0.55/mutated: 0.62). 

MAPP-MMR tool classified it as neutral with a score equal to 2.130.  In brief, by using other online supplementary prediction tools (DANN, FATHMM v2.3, MutationAssessor, LRT…), this variant has been described as “deleterious or pathogenic” by 13 out of 15 tested tools.

The discussion section is way too extensive. It is informative, no doubt, but maybe it’s better to shorten it to improve readability. Kindly consider. 

The discussion section has been shortened a bit as recommended: 

-The paragraphs (from lines from 484 to 456; lines from 498 to 506; lines from 555 to 561),  have been removed in order to reduce the discussion section as recommended. 

-The paragraph from lines 612 to 629  has been resumed.

It is important to highlight that for a subsequent classification of the MSH2 variant, it needs to be assessed by functional studies either in vitro (PMID: 30504929) or in human cells (PMID: 31237724). 

Regarding the MSH2 variant assessment by functional studies either in vitro (PMID: 30504929) or in human cells (PMID: 31237724),we added this point at the end of the conclusion section (lines from 691 to 693) as perspective to this work.

Round 2

Reviewer 1 Report

This manuscript describes that germline MSH2 p.R243Q variant is involved in Lynch Syndrome II. The presented manuscript is revised adequately.